# Immune Regulation by Dendritic Cell Extracellular Vesicles in Cancer Immunotherapy and Vaccines

**DOI:** 10.3390/cancers12123558

**Published:** 2020-11-28

**Authors:** Irene Fernández-Delgado, Diego Calzada-Fraile, Francisco Sánchez-Madrid

**Affiliations:** 1Vascular Pathophysiology Department, Centro Nacional de Investigaciones Cardiovasculares (CNIC), 28029 Madrid, Spain; irene.fernandez@cnic.es (I.F.-D.); diego.calzada@cnic.es (D.C.-F.); 2Immunology Service, Health Research Institute Hospital La Princesa (IIS-IP), Faculty of Medicine, Universidad Autónoma de Madrid (UAM), 28006 Madrid, Spain; 3Center for Biomedical Research Network on Cardiovascular Diseases (CIBERCV), Carlos III Health Institute, 28029 Madrid, Spain

**Keywords:** extracellular vesicles (EVs), dendritic cell (DC), cancer, immunotherapy, vaccines, tumor-derived EVs, oncopathogens

## Abstract

**Simple Summary:**

Dendritic cells have a central role in starting and regulating immune functions in anticancer responses. The crosstalk of dendritic cells with tumors and other immune cell subsets is partly mediated by extracellular vesicles (EVs) secreted by both cell types and is multidirectional. In the case of dendritic cell EVs, the presence of stimulatory molecules and their ability to promote tumor antigen-specific responses, have raised interest in their uses as therapeutics vehicles. In this review, we highlight how dendritic cell- and tumor cell-derived EVs affect antitumor immune responses. In addition, we discuss the different approaches that exploit dendritic cell EVs as a novel platform for immunotherapies and therapeutic and prophylactic anticancer vaccines.

**Abstract:**

Extracellular vesicles (EVs) play a crucial role in intercellular communication as vehicles for the transport of membrane and cytosolic proteins, lipids, and nucleic acids including different RNAs. Dendritic cells (DCs)-derived EVs (DEVs), albeit variably, express major histocompatibility complex (MHC)-peptide complexes and co-stimulatory molecules on their surface that enable the interaction with other immune cells such as CD8^+^ T cells, and other ligands that stimulate natural killer (NK) cells, thereby instructing tumor rejection, and counteracting immune-suppressive tumor microenvironment. Malignant cells oppose this effect by secreting EVs bearing a variety of molecules that block DCs function. For instance, tumor-derived EVs (TDEVs) can impair myeloid cell differentiation resulting in myeloid-derived suppressor cells (MDSCs) generation. Hence, the unique composition of EVs makes them suitable candidates for the development of new cancer treatment approaches including prophylactic vaccine targeting oncogenic pathogens, cancer vaccines, and cancer immunotherapeutics. We offer a perspective from both cell sides, DCs, and tumor cells, on how EVs regulate the antitumor immune response, and how this translates into promising therapeutic options by reviewing the latest advancement in DEV-based cancer therapeutics.

## 1. Introduction

Cancer is a very heterogeneous disease that can develop in almost any tissue due to the tumorigenic transformation of normal cells. Malignant cell transformation is a multi-step, diverse process that might be instigated by genetic factors. Hanahan and Weinberg outlined the six core “hallmarks of cancer” which are: sustained proliferative signaling, evasion of growth suppressors, activation of invasion and metastasis, replicative immortality, angiogenesis, and resistance to cell death; with two emerging hallmarks: metabolic deregulation and immune evasion; and two enabling characteristics that facilitate the process: genome instability and mutation, and tumor-promotion of inflammation [1]. These hallmarks place the spotlight on the role of the tumor microenvironment in malignant cell progression. 

Cancer thrives after escaping control checkpoints. The immune system exerts one of the main defense mechanisms against malignant cells via immune surveillance and specific antitumor immune responses. This comprises two arms: innate immune cells that define a first rapid line of defense and adaptive immune cells, which drive an antigen-specific response that includes the development of long-term memory. However, memory traits have also been observed in the innate compartment based on epigenetic and metabolic reprogramming of cells, the so-called “trained immunity” [2]. Among innate immune cells, the main players are phagocytic cells such as neutrophils, monocytes, and macrophages; professional antigen presenting cells (APCs) such as dendritic cells (DCs), or cell slayers such as natural killers (NKs). Adaptive immune cells include antibody (Ab)-producers B lymphocytes; CD8^+^ cytotoxic T cells; or helper cells, which include the family of CD4^+^ T cells (Th_1_, Th_2_, Th_17_, T_reg_, and others). Around and within tumor environment, innate and adaptive immune cells are crucial players [3]. 

Immature DCs switch to an activated state through a maturation process after stimulation by a danger signal, such as sensing a pathogen-derived molecule, or tissue damage [4]. These cells infiltrate the tumor and recruit effector cells. DCs represent the most effective APCs able to prime naïve T cells and induce an effective antigen-specific antitumor defense. They include a vast variety of cellular types with diverse functions depending on their origin, location, and properties. For instance, DCs can be subdivided into: (i) conventional DCs (cDCs), either resident of lymphoid tissues or migratory, where we can find cDC1 required more for pathogen/tumor immune responses or cDC2 more focus on major histocompatibility complex (MHC)-II based responses; (ii) plasmacytoid DCs, main producers of type 1 interferon (IFN), (iii) tissue-specific DCs such as Langerhans cells (LCs) or dermal DCs (dDCs) and (iv) monocytic-derived DCs (moDCs), producers of tumor necrosis factor (TNF)-α and inducible nitric oxide synthase (iNOS) [5]. In particular, cDC1 are critical in tumor surveillance, antitumor antigen-specific T cell responses, responsiveness to immunotherapies, and are associated with increased patient survival [6]. 

Extracellular vesicles (EVs) are secreted by cells to the extracellular milieu [7] and comprise mainly three subgroups depending on their origin: (i) shedding vesicles, generated by evagination of the plasma membrane; (ii) exosomes, generated at the multivesicular bodies and secreted by its fusion with the plasma membrane; and (iii) apoptotic vesicles [8,9,10]. These vesicles are constituted by a lipid bilayer containing an assortment of proteins, lipids, metabolites, and nucleic acids, the latter including microRNA (miRNA), mRNA, long non-coding RNA (lncRNA), DNA and mitochondrial DNA (mitDNA) [10,11,12,13]. Although their biogenesis is still under study, there is evidence of an active sorting process of molecules into these vesicles as its cargo is not a mere reflection of the cell content [10,14,15]. For instance, sorting mechanisms include tetraspanins, lipids, specific proteins, post-translational modifications, or endosomal sorting complexes required for transport (ESCRT)-dependent processes [14,16,17,18,19]. Their functions include a variety of cellular processes, but these vesicles are specialized in intercellular communication [20,21]. EVs can function as autocrine, paracrine, or endocrine entering into circulation. Once they reach their final destination, the mechanisms underlying their internalization and signaling processes remain still under consideration [7,8,10]. Almost any type of cell can secrete EVs, including malignant cells and immune cells [8,10,11,21]. There are specific markers for each type of vesicle, specific for a certain cell type or even to distinguish between types of vesicles from the same cell type [21,22,23,24]. Besides, EVs can be detected in any type of biological fluid [21,22]. For example, as EVs from malignant cells convey tumor molecules, they represent good tumor biomarkers and excellent liquid tumor biopsies [25,26]. Interestingly, in cancer patients, the amount of serum-EVs were shown to correlate with a poor prognosis [27]. Vesicle secretion and size heterogeneity has made it difficult to decipher their precise origin and functions, generating some controversy [28,29,30,31,32]. 

In this review, we will highlight the intercellular communication through EVs between tumor cells and immune cells, especially DCs. DCs actively produce EVs to enhance and modulate the immune response, while tumor cells also secrete EVs in order to counteract this process. These characteristics make EVs a very promising tool for the development of immunotherapies. Hence, we will also discuss the latest advancements in vaccination and immunotherapies strategies based on EVs to stop or slow down tumor progression and metastasis.

## 2. The Role of EVs in Cancer Development

### 2.1. Modulation of Antitumor Immunity by Dendritic Cells (DCs)-Derived Extracellular Vesicles (EVs)

#### 2.1.1. DCs-Derived EVs (DEVs)

Immune cells can secrete immunologically active EVs [8]. One of the first studies of vesicles in immune responses described the role of B cell-EVs carrying MHC-II molecules on their surface in driving T cell proliferation [33]. Since then, the number of publications on immune cell-derived vesicles functions has not ceased to grow due to their potential in human immunotherapies, for instance against cancer. Among innate immune cells, we will focus our attention on EVs derived from DCs (DEVs) [34]. At the end of the 90s, Zitvogel and colleagues showed that DEVs convey tumor-associated antigens (TAAs) promoting antitumor immunity by effector T cells [35]. In particular, DEVs contain a specific repertoire of molecules like T cell co-stimulatory molecules (CD86, CD80, CD40) [36,37], antigen presenting molecules (MHC-II, MHC-I) [36,37,38], adhesion molecules (Integrins, intercellular adhesion molecule 1 (ICAM-1), dendritic cell-specific intercellular adhesion molecule-3-Grabbing non-integrin (DC-SIGN)) [39], NK modulation molecules (TNF-α, interleukin 15 receptor α (IL-15Rα), NKG2D-L) [40,41] and the EV markers such as tetraspanins (CD9, CD81, CD63), ESCRT complex proteins (Tumor susceptibility gene 101 (TSG101), ALG-2-interacting protein X (ALIX)), heat shock proteins (HSC73, HSP84) and others (SYNTENIN-1, ACTIN) [37]. The amount of MHC molecules or co-stimulatory molecules depends on the physiological state of the DC [38,42,43]. In fact, EVs secreted from DCs can transfer functional MHC-I-peptide complexes to other DCs [44]. Not only DEVs induce stimulation of naïve CD4^+^ T cells, but also these EVs are also used by mature DCs as a source of tumor antigens [43,45]. Apart from proteins, nucleic acids sorted within DEVs play an important role in the regulation of immune responses, in particular miRNAs both from immature and mature DCs [46,47,48]. The secretion of these vesicles by DCs can be altered upon exposure to different stimuli [49,50]. Highlighting the heterogeneous nature of these vesicles, different subtypes of DEVs could also be found, each of which could perform a variety of functions [50,51]. 

#### 2.1.2. DEVs Function

One of the main functions of DEVs is T cell activation (Figure 1A). EVs from an antigen-loaded DC bearing tumor antigens that may promote CD4^+^ and CD8^+^ T cell responses by direct antigen presentation, which leads to tumor growth suppression [35,36,52] with increased efficiency in the case of mature DC-derived EVs [39]. DEVs can also be internalized by other DCs as a source of exogenous peptide-loaded MHC (pMHC), which may be subsequently presented to naïve, primed, or memory T cells [43,44,45,53,54,55]. This process has a special relevance in organ transplantation as acceptor DCs incorporate donor-DEVs to stimulate allospecific T cells [56]. Moreover, by coating DCs with the pMHC-loaded EVs, a process known as MHC cross-dressing, T cell response is reinforced and amplified [57,58]. In addition, DCs can incorporate pMHC-loaded vesicles from other cell origins, such as epithelial cells, to potentiate antigen presentation to T cells [59]. Also contributing to the global immune response, activated T cells can recruit pMHC-loaded EVs on a leukocyte function-associated antigen-1 (LFA-1)-dependent manner during antigen-specific immune synapse (IS) formation [38,60], where activated B cells also secrete a portion of these EVs [61]. Another mechanism and source of tumor antigens is the internalization of EVs containing the full antigen or peptide, that would be presented later by endogenous MHC-I molecules at the acceptor cell in a process called cross-presentation [62]. In fact, EVs loaded with the whole antigen are more efficient in antigen presentation than EVs bearing only pMHC [63,64]. These antigen-loaded EVs elicit a Th1 CD4^+^ and CD8^+^ T cell response dependent on B cell activation [65,66]. This response might also be enhanced by the presence of CD80 and ICAM-1 in those EVs [39]. Besides, bystander T cells can promote DC maturation in the absence of innate stimuli [67,68,69], which is also reflected in DEVs supporting subsequent specific antigen T cell activation [70]. 

Antigen-loaded DEVs can play many other functions in addition to modulating T cell responses, such as promoting humoral immunity [71,72]. DEVs also contain NKG2D-L and IL-15Rα, contributing to NK cell activation and proliferation [41]. DEVs can directly activate NK cells via TNF-α as they display it on their surface together with FasL and TRAIL. These NK cell responses, along with the apoptotic signaling, contribute to tumor cell removal [73]. Furthermore, EVs from heat shocked-activated DCs bear BAT3, the ligand for NKp30, mediating cytokine release and NK cell cytotoxicity [74] (Figure 1A). Therefore, DEVs entail an important tool to elicit the immune response against malignant cells. These vesicles can boost both T cell and NK responses, whose cytotoxicity leads to tumor cell killing [34,40]. Other DCs can internalize or be coated with DEVs enhancing the antitumor response [43,58]. 

#### 2.1.3. Influence of the Adaptive Response 

On the other side of an antigen-specific IS, T cells can also produce EVs (TEVs) [11]. In fact, during IS, TEVs can transfer miRNA to APCs [75,76] as B lymphocytes, thus affecting germinal center reaction and Ab production [77]. In addition, during the IS, TEVs bearing DNA and mitDNA can prime DCs against subsequent viral infections [78]. Furthermore, regulatory T cells (T_reg_), which are a T cell subtype that maintains self-tolerance and restrains other immune responses, can also affect DC function. EVs from T_reg_ cells modify DC function by miRNA transfer, inducing a tolerogenic phenotype accompanied by increased interleukin (IL)-10 and decreased IL-6 secretion [79]. Macrophage-mediated immune suppression is also regulated by EVs from antigen-specific CD8^+^ T suppressor cells (Ts) through the induction of T_reg_ and inhibition of T cell effector proliferation [80]. Therefore, EVs at the tumor microenvironment are crucial to delineate the final outcome of the disease, leading either to a pro-tumorigenic or antitumorigenic situation through immunomodulation. 

### 2.2. Modulation of the Immune Response by Tumor-Derived EVs

#### 2.2.1. Tumor-Derived EVs (TDEVs)

Immune evasion is one of the emerging hallmarks for cancer [1]. Tumor cells manage to escape immune responses, particularly inhibiting T cell activation and DC differentiation [81]. Notably, tumor cells can shed a large amount of EVs, highlighting the importance of vesicle secretion during tumor development [82,83]. Tumor-derived EVs (TDEVs) can mediate many aspects of the immune response [84]. TDEVs can alter the microenvironment of the tumor promoting both pro/anti-inflammatory responses on monocytes, macrophages, and DCs, as well as anti-inflammatory responses acting on NKs and T_reg_, thus modulating angiogenesis, invasion, apoptosis, and metastasis [84,85]. TDEVs not only contain the typical exosomal markers, like tetraspanins (CD81, CD63), ESCRT proteins (TSG101), heat shock proteins (HSC70), but also tumor-specific markers such as TAAs (Her2/Neu, Mart1, TRP, GP100), MHC molecules, TNF, Fas Ligand, CD73 or Galectin-9 [82,86]. One of the mechanisms of immune cell evasion is the expression of the programmed death-ligand 1 (PD-L1) on the malignant cell surface. This ligand binds the receptor programmed cell death protein (PD)-1 on CD8^+^ T cells, thereby inhibiting their effector functions and leading to so-called adaptive immune resistance [87]. Interestingly, PD-L1 is also present in TDEVs suppressing CD8^+^ cytotoxicity and facilitating tumor growth. The number of PD-L1 vesicles increases with interferon (IFN)-γ stimulation [88].

#### 2.2.2. TDEVs Function on Myeloid Cells

TDEVs convey several molecules to modulate the activity of the surrounding myeloid cells. For instance, myeloid-derived suppressor cells (MDSCs), which comprise a heterogeneous population of immature myeloid cells with T suppressive abilities, are accumulated around the tumor through the effect of TDEVs on bone marrow cells [89,90]. EVs can affect monocytic cell function, which is Toll-Like Receptor (TLR)-dependent, by triggering a signaling cascade that activates the nuclear factor κB (NFκB) and signal transducer and activator of transcription 3 (STAT3). This signal triggers the production of the pro-inflammatory cytokines IL-6, IL-1β, IL-8 and TNF-α [91]. NFκB renders an important function in malignant tumor cells as it does not only play a part in immunity but also regulates cell proliferation, apoptosis, and cell migration [92]. Furthermore, TDEVs from chronic lymphocytic leukemia (CLL) bear a small non-coding RNA (hY4). The hY4 RNA apparently binds to TLR7, inducing PD-L1 expression and the release of various cytokines by monocytes, such as C-C motif chemokine ligand (CCL) 2, CCL4, and IL-6. This leads to a pro-tumorigenic phenotype favoring the progression of the tumor [93]. Like monocytes, TDEVs loaded with small RNAs (miR-21 and miR-29a) promote an inflammatory response via TLRs-NFκB driven by macrophages, and promote tumor growth and metastasis [94]. Additionally, macrophages can produce inflammatory cytokines, including IL-6, IL-8, IL-1β, CCL2, TNF-α and granulocyte colony-stimulating factor (G-CSF), also via NFκB and TLR-dependent by TDEVs stimulation in breast [95] and gastric cancer [96]; the later with increasing tumor cell growth, migration and invasion [96]. Furthermore, M2-polarization, which is more pro-tumorigenic, can be controlled by EVs derived from hepatocellular carcinoma (HCC) cells through SALL4/miRNA146a axis [97]. These TDEVs also contain lncRNA TUC339 contributing to M1/M2-polarization of macrophages [98].

#### 2.2.3. TDEVs Function on DCs

“Normal” DCs pose a great threat to the progression of the tumor [99]. Therefore, it is not surprising that tumor cells contain a repertoire of mechanisms to inhibit their function, including TDEVs [100] (Figure 1B). For instance, it has been described that TDEVs from melanoma or colorectal carcinoma cells impair CD14^+^ human monocyte differentiation into DCs, leading to MDSCs generation creating an immunotolerant environment. This process was accompanied by suppressive activity on T cell proliferation and the release of higher amounts of IL-6, TNF-α, and transforming growth factor (TGF)-β [101]. In another study, EVs from mammary tumor cells target CD11b^+^ myeloid precursors in the bone marrow, also inhibiting their differentiation into DCs, where IL-6 plays a key role in this blockade [89]. These effects on DCs differentiation blockade have also been observed in other types of tumors, like renal carcinoma, Lewis lung carcinoma (LLC) or breast cancer [102,103]. For instance, TDEVs from lung and breast cancer also block the differentiation of DCs precursor cells, inducing cell apoptosis and leading to a drastic decrease of Th_1_ T cell differentiation. This effect could be partially reverted, blocking PD-L1 [102]. Likewise, EVs released by renal carcinoma stem cells CD105^+^, which express the non-classical human leukocyte antigen (HLA)-G, interfered with DCs ability to maturate and differentiate from monocytes, counteracting their ability to stimulate T cells [103]. These TDEVs are able to modify the tumor microenvironment by inducing angiogenesis and travelling to other organs, such as lungs, inducing the formation of a pre-metastatic niche [104]. HLA-G, -E and -F overexpression constitutes an immunosuppressive mechanism, as it hampers the cytolytic activity of effector cells, thus allowing malignant cells to escape immunosurveillance [105]. Specially, HLA-G has been shown to influence DCs to behave as regulatory DCs inducing tolerance [106] and play a key role in cancer and immune evasion [107]. Moreover, EVs secreted from melanoma cells also bear HLA-G [108]. Recently, it has been described that tumor microenvironment, including TDEVs, educate DCs to promote tumorigenesis and metastasis. TDEVs carry heat-shock proteins such as HSP72 and HSP105 on their surface, which bind TLRs in DCs and lead to IL-6 secretion, which in turn promotes tumor metastasis via matrix metalloproteinase (MMP) 9 [109]. Similar to monocytes, miRNAs in TDEVs also affect DCs function. Treatment with TDEVs from pancreatic cancer cells bearing miR-203 downregulate TLR4 and downstream cytokines (TNF-α and IL-12) in DCs [110]. In contrast, TDEVs bearing TAAs might support immune responses against the tumor by increasing tumor antigen-presentation and T cell response [111]. TDEVs convey miR-155 that can promote DC maturation thereby resulting in the activation of CD8 cell antitumor response [112,113]. However, immunosuppression seems to be the dominant effect of TDEVs over antigen cross-presentation, at least in prostate cancer [114]. Interestingly, TDEVs from irradiated breast cancer cells transfer tumor double-stranded DNA to DCs and stimulate IFN type I activation via stimulator of interferon genes (STING) [115]. Hence, TDEVs contain a repertoire of proteins, DNA and miRNAs that alter DC function and differentiation, and lead to a change in the immune response, thus favoring or hampering tumor progression. Therefore, the design of DC-based immunotherapies must take into account the conundrum that DC per se might fight the tumor, but TDEVs at the tumor microenvironment might sway DCs function into a pro-tumor phenotype.

#### 2.2.4. TDEVs Function on Other Immune Cells

As well as innate responses, adaptive immune cells fight against malignant cancer cells. Thus, tumor cells and TDEVs actively counteract those cells’ functions. For instance, vesicles from pancreatic ductal adenocarcinoma (PDAC) carry TAAs that induce B cell production of auto-Abs acting as decoys for complement-mediated cytotoxicity, originally directed towards malignant cells [116]. Besides, TDEVs decoy function can control NK cell cytotoxicity as they carry NKG2D ligands on their surface [117,118]. As we have mentioned, DCs are specialized in T cell activation leading to an adaptive immune response against the malignant cells. Therefore, any effect by TDEVs on DCs affects T cell function. Nevertheless, TDEVs can also directly affect T cell function. For instance, TDEVs from Epstein-Barr virus (EBV)-associated nasopharyngeal carcinoma (NPC) carry Galectin-9 and induce T cell anergy and death through its binding to the receptor TIM3 [119].

Tumors have managed to contain a repertoire of mechanisms to evade any antitumoral response. In particular, the importance of TDEVs on these processes is remarkable. TDEVs can hinder many aspects of antitumor immune response, from increasing suppressor cell populations, like MDSCs, to decreasing antigen presentation processes. On the contrary, DCs can take advantage of TDEVs as they convey TAAs, which might serve as a source for direct or indirect presentation mechanisms. However, this remains to be fully characterized, especially in the case of direct TAAs presentation carried by DEVs. Finally, diversity in EVs methodology and dendritic cell sources used in these studies may pose a confounding factor when interpreting these data.

## 3. DEV-Based Cancer Therapeutics

The fine modulation of the immune responses that EVs are able to perform, as well as their abilities for shuttling different biomolecules, including proteins that may serve as antigens to mount an immune response, have made EVs interesting candidates for different uses in therapeutic and prevention contexts where the immune system has a major role. Given the importance of EVs derived from immune cells in immune modulation, and specifically of DCs as major coordinators of immune responses, studies using DEVs pioneered the field of the application of EVs with a therapeutic purpose and have expanded from the field of cancer [120,121] to vaccines for infectious diseases [122] and other immune-mediated conditions affecting the nervous system [123,124,125,126].

A pioneering study using autologous DEVs showed rejection of tumors in mice when loaded with tumor peptides and therefore used as a cell-free vaccine [35]. Driven by these observations, several in vivo and clinical trials have followed and explored the use of EVs (more prominently DEVs) as potential immunotherapeutic agents in cancer (reviewed in [120,121]). These therapeutic agents possess several advantages over DC-based vaccines. First, the molecular composition of DEVs is more restricted and controllable than that of whole cells, owing to specific sorting and loading mechanisms. Also, DCs shelf life is more limited compared to DEVs. Regarding their delivery, DEVs can be engineered for a targeted delivery of their content and can easily reach the proper location on secondary lymphoid organs compared to the chemokine-dependent migration of DCs [121]. Besides, DCs are susceptible to immunosuppression exerted by checkpoint signaling or immunosuppressive immune cells such as T_regs_ and MDSCs, whereas DEVs are not. Moreover, DEVs possess 10–100 times more pMHC complexes per surface than DCs [127] and are enriched in NK cell activation ligands [41]. Nonetheless, no DEV-based therapy has been approved to date whilst two DC-based vaccines have received marketing authorization: Sipuleucel-T (Provenge^®^), approved by the US Food and Drug Administration in 2010; and APCEDEN^®^, which received authorization by the Indian Central Drugs Standard Control Organization in 2017. Still, these two DC-based vaccines show limited clinical benefit [128]. 

In this section, we will focus on the use of DEVs as therapeutic agents in the context of cancer. Importantly, we differentiate between vaccination approaches, when EVs are loaded with tumor antigens in order to elicit (tumor) antigen-specific immune responses, or immunotherapies if they are not loaded with tumor antigens or their therapeutic effects are based on immunomodulation independently on whether they improve a later antitumoral antigen-specific immune response.

### 3.1. DEV-Based Vaccines

Several sources of EVs have been used as potential tumor vaccine candidates. Importantly, TDEVs have raised a great interest because they are a primary source of tumor antigens that are potentially preloaded with peptides representative of the tumor mutanome, and they are as well able to stimulate antitumor DC responses [129]. Also, alternative sources of EVs, e.g., other myeloid cell types, have been explored [85]. However, DEVs are the most interesting EVs to activate specific immune responses because: (1) they are antigen-presenting platforms due to the fact that they contain functional molecules that have a major role in the IS such as MHC-I and-II or CD1 loaded with tumor peptides as well as co-stimulatory molecules (CD80, CD86) and molecules that enhance cell adhesion (ICAM-1) ([37,39,130,131] reviewed in [132]); and (2) because the activation state of their cellular source can be manipulated so that they are devoid of potentially immunosuppressive molecules that may be present in TDEVs [88]. The induction of antigen-specific responses by DEVs has been demonstrated by the activation of tumor-specific CD4^+^ and CD8^+^ T cells via engagement of their TCR and is superior to the induction of T cell responses by TDEVs [133,134]. This mechanism, rather than relying on direct activation of T cells by the DEV-loaded pMHC complexes, is dependent on the transfer of these complexes to different host DC subsets, increasing the number of DCs carrying tumor pMHC complexes and acting as an amplifier of the adaptive response [44,45,53]. Moreover, EVs also carry cytokines [135] and, although this may help to tailor the T cell response, it may not have a major role in DEVs as the pMHC recipient DCs were shown to be already mature and activated (CD80^+^ or CD86^+^) [45]. Other antitumor mechanisms of DEVs include the activation of NK cells, which, although not antigen-specific, may mediate directly or indirectly the enhancement of T cell-specific responses by these or other cellular subsets as it has been shown for some adjuvanted vaccines [136].

#### 3.1.1. DEVs as Tumor Vaccines

Due to the feasibility of large production of DCs for DEVs generation, most studies that explored the use of DEVs in antitumor settings in ex vivo, in vitro and in vivo have used murine bone-marrow-derived DCs (BMDCs) as a source for DEVs production. These DEVs have been loaded with different TAAs to mount specific T cell responses against tumors (Figure 2A). These models include for example the loading of α-fetoprotein in HCC models [137,138] or chaperone-rich lysates from the GL261 glioma cell line in glioma models [139]. In these cases, antigen-loaded DEVs were able to increase overall survival and reduce tumor growth. These effects were accompanied by a potent activation of both CD4^+^ and CD8^+^ T cells responses as measured by IFNγ and IL-2 production, their infiltration in the tumor, and a reshaping of the immune tumor microenvironment that included reduction of T_reg_ numbers and less IL-10 and TGF-β production. Importantly, two models of DEV loading with whole ovalbumin (OVA) protein have shown that whole-protein-loaded DEVs are able to induce both humoral and cellular immune responses against B16-OVA tumors [66,140]. Indeed, one of these studies demonstrated increased antitumor efficacy of DEVs loaded with whole OVA protein compared to DEVs loaded with the MHC-I-restricted OVA_257–264_ peptide [66].

Evidence has pointed out that the use of activated DC as producers of DEVs increases the efficacy of DEVs when used as cancer vaccines [140]. Owing to this, several studies have explored the use of DEVs coming from DCs stimulated with TLR ligands. For instance, polyinosinic:polycytidylic acid (poly(I:C)) treatment of BMDCs increased the antitumor efficacy of DEVs loaded with the human papillomavirus (HPV) peptide E7_49–57_ both in vitro with TC-1 cells (transformed with E7 protein) and in a TC-1 model of cervical cancer in vivo [141]. In both settings, poly(I:C) DEVs activated antigen-specific CD8^+^ T cells measured by IFNγ production and resulted in a prolonged survival rate in vivo. In order to decipher which TLR stimulation may be more efficient, a study compared the antitumor efficacy of DEVs produced from DC stimulated with poly(I:C), lipopolysaccharide (LPS) and CpG (ligands of TLR3, TLR4 and TLR9 respectively). OVA-loaded DEVs of the TLR3 stimulated group were better stimulators of OT-I CD8^+^ and OT-II CD4^+^ T cells in vivo. In accordance, mice vaccinated with DEVs from the poly(I:C) group had less tumor growth and longer survival in an in vivo B16-F10 tumor model, correlating with more robust activation of tumor-specific CD8^+^ T cells and recruitment NK and NKT cells [142].

Few ex vivo studies have used human DCs as a source of DEVs. Still, a study used DEVs produced by human cord-blood-derived DCs and checked their antitumoral capacities. Contrary to loading DEVs with tumor peptides, total tumor RNA from a human gastric adenocarcinoma cell line (BGC823) was loaded into DEVs. These RNA-loaded DEVs were superior in inducing tumor-specific cytotoxic T lymphocytes (CTL) proliferation compared to tumor-lysate-loaded DEVs ex vivo. Importantly, this study provided an alternate to moDCs as a source for large production of human DEVs for antitumor vaccination approaches [143]. 

#### 3.1.2. DEVs as Vaccines for Oncogenic Pathogens

In all these cases, DEVs are used as therapeutic rather than prophylactic vaccines. Indeed, the prophylactic use of DEVs against cancer is largely unexplored because intervention is usually used when the disease appears. However, many pathogens have been described to induce carcinogenesis by direct or indirect mechanisms. These pathogens include viruses (including HPV, hepatitis B virus (HBV), hepatitis C virus (HCV), human T lymphotropic virus (HTLV) or human immunodeficiency virus (HIV)-1) and bacteria (*Helicobacter pylori*) [144]. Approximately 2.2 million cases of cancer attributable to these infections were diagnosed in 2018 [144]. Other pathogens described to be associated with human cancer include parasitic diseases such protozoa (*Plasmodium falciparum*), and trematodes (*Schistosoma haematobium*) [145]. Hence, implementation of prophylactic vaccines that target oncopathogens constitute a potential health benefit. For example, implementation of HPV vaccination programs has successfully lowered the burden of cervical cancer [146]. However, except for HPV and HBV, effective vaccines against oncogenic pathogens are lacking.

EVs, in general, have been explored as a novel vaccine platform for infectious diseases to a lower extent than cancer vaccines, and no clinical trials have been performed to date. In some cases, EVs have shown increased efficacy than traditional vaccine formulations against pathogens. Targeting of pathogen proteins to EVs endowed efficient antigen-specific cellular [147] and humoral immune responses [148] (Figure 2B). Despite the great potential of the EV-based vaccine platforms, a limited number of studies have been performed using oncogenic pathogens as therapeutic targets. However, the versatility of these platforms and their similarities with other efficacious and related vesicle-based platforms such as bacterial outer membrane vesicles [149] make EVs a promising vaccine platform for targeting oncogenic pathogens.

DEVs have also been explored to generate protection against toxoplasmosis [71,150]. Alternatively, the immunostimulatory capacities of DEVs have been explored as vaccine adjuvants or adjuvant carriers. DEVs are able to carry TLR ligands that can stimulate bystander DCs, ultimately inducing IFNγ production and NK cell activation [151]. Indeed, TLR3 and TLR9 ligands have combined to DEVs and have been shown to be superior inducers of antigen-specific CD8^+^ T cell responses [152]. Also, DEVs derived from an LPS-stimulated monocyte cell line administered subcutaneously were able to potentiate the cellular immune response to the surface antigen of the HBV (HBsAg) when compared to HBsAg formulated without these vesicles, thereby preventing infection by the oncogenic HBV [153]. Lastly, other cellular EVs sources have been explored and described to elicit antigen-specific antipathogen responses such as T cells [154,155] or infected cells [156,157,158,159].

### 3.2. DEV-Based Immunotherapies

Using human moDCs, Tkach et al. described that DEVs stimulated CD4^+^ T cell ex vivo. Activation was biased towards generating a Th1-type response when using small DEVs; and towards Th2 when using large DEVs from immature cells. Both DEVs sizes induced Th1 cytokines when DCs were matured with IFNγ [50]. As DCs were not loaded with a specific antigen and a pool of CD4^+^ T cells, these results provide an example of DEVs modulating T cell activation. After the observation in a phase I clinical trial (discussed later) that DEVs pulsed with tumor peptides were unable to mount a T cell-specific response but induced NK cell activation, another study explored the role of DEV-mediated NK cell activation as we mentioned in a previous section. Using murine BMDC DEVs administered intradermally to mice, they observed NK cell proliferation and activation that was mediated by IL-15Rα and NKG2D ligands present in DEVs (Figure 2C). Also, DEVs derived from human moDCs induced NK cell activation was dependent on NKG2D ligands [41]. Besides, Munich and colleagues used murine BMDC DEVs loaded with TNF, FasL and TRAIL describing that these vesicles had the ability to directly kill B16 tumor cells ex vivo by promoting apoptosis, and also induced NK cell activation and their IFNγ production [73]. As previously discussed, loading TLR ligands to DEVs, NK cell activation and bystander DC activation can be achieved [151]. Further exploring the antitumor therapeutic effects of innate cell activation, Gehrmann U. et al. used murine BMDC DEVs loaded with OVA protein and the NKT cell activator α-galactosylceramide (αGalCer) in an in vivo B16-OVA tumor model. Besides inducing T and B cell-specific responses, DEVs loaded with αGalCer were able to induce a strong innate response based on NKT and γδ T cell activation that resulted in tumor CD8^+^ T cell infiltration, reduced tumor growth, and increased survival [160]. DEVs have also been used for direct drug delivery. DEVs from an immature DC cell line loaded with doxorubicin selectively targeted cells expressing αv integrin via exosomal loading of the exosomal membrane protein lysosome-associated membrane protein 2 (LAMP2b) fused to the corresponding integrin-specific peptide iRGD. This allowed for efficient targeting of breast cancer cells in vitro, tumor targeting in vivo, and reduced tumor growth after their intravenous injection [161]. Besides, DEVs from DCs exposed to stress conditions as heat and high concentration of CO_2_ were able to inhibit a gastric cancer cell line proliferation via apoptosis induction [162].

### 3.3. Clinical Trials Using DEVs

To date, four phase I and one phase II clinical trials have been performed using DEVs as immunotherapeutic agents (summarized in Table 1). Two phase I trials pioneered the use of DEVs in 2005. Morse MA. et al. used moDC-derived EVs loaded with both MHC-I and -II melanoma antigen gene (MAGE) peptides directly in isolated DEVs, or indirectly loading cultured autologous moDCs. An MHC-I-restricted cytomegalovirus (CMV) peptide and an MHC-II-restricted tetanus toxoid-derived peptide were also loaded in the DEVs. DEVs were injected once per week along 4 weeks to 9 patients that completed therapy. Delayed-type hypersensitivity (DTH) responses were detected in 3 subjects and robust CMV responses were detected in 3 individuals, whereas no MAGE T cell responses were observed. Increase of T_regs_ and partial enhancement of NK cell functions were observed in some individuals [163]. In the other pioneering clinical trial, Escudier B. et al. administered DEVs produced by moDCs on 15 metastatic melanoma patients. Here, DEVs were used as a vaccine platform as they were loaded with MAGE3 MHC-II and MCH-I peptides (MAGE3_168–176_ and MAGE3_247–258_ respectively) and administered four times, once per week. Specific CTL responses in peripheral blood were not induced or clonally expanded, and only tumor infiltration of activated T cells was described for a partial response case. On the contrary, NK cell effector functions were shown to be increased [164]. As NK cell functions were partly activated in both studies and, as previously discussed, NK cell activation and proliferation could be induced by DEVs via DEV-loaded NKG2D and NKp30 ligands, and IL-15Rα [41,74]; Viaud S. et al. further explored whether the NK cell activation could account for the observed clinical effects despite the lack of a T cell response in the clinical study performed by Escudier B. et al. They described that NK cell numbers were increased in peripheral blood and NKG2D expression levels were restored both on T and NK cells, enhancing the NKG2D-dependent cytotoxicity [41]. Another clinical trial phase I by Dai S. et al. used ascites-derived EVs (AEVs) in combination with granulocyte macrophage colony-stimulating factor (GM-CSF) to treat 40 patients with advanced colorectal cancer. These vesicles, similar to DEVs, were enriched not only in MHC-I and MHC-II, but also in CD80 or ICAM-1, and possessed the carcinoembryonic antigen (CEA) as they were derived from patient malignant ascites. A DTH response as well as a CEA-specific CTL cell response was efficiently induced when combining AEVs with GM-CSF [165].

Due to the limited clinical benefit shown by these trials, which used immature DC for DEVs production, Besse B. et al. conducted a Phase II clinical trial using DEVs derived from IFNγ-matured DC [121]. In this trial, 22 non-small-cell lung carcinoma (NSCLC) patients were administered 4 times with these DEVs loaded with tumor peptides as a maintenance treatment after chemotherapy. Although no tumor antigen-specific T cell responses could be detected and the primary endpoint of the study was not reached (50% progression-free survival), the therapy induced a stabilization of the progression in 32% of patients and one patient benefited from long-term stabilization. These clinical effects in the absence of T cell responses were probably due to the observed increase in NK cell functions. These DEVs contained BAG6, a ligand for NKp30 which is induced by IFNγ DC maturation, which correlated with greater NKp30-dependent production of TNFα and IFNγ by NK cells [166].

Importantly, although in many cases only partial or minor responses were observed, some patients exhibited stability of the disease progression. Besides, these clinical trials have proven DEVs to be well-tolerated and safe, and amenable to be produced in large-scale set ups.

### 3.4. The Future of DEVs in Vaccination Approaches

The increased efficacy and versatility of EV-based vaccines have made them potential candidates for rapid development of vaccines against emerging infections [122,167]. For example, EVs targeting has not only increased immunogenicity of EV-based vaccines as discussed before, but it has also been shown to improve the humoral responses in adenoviral vector vaccines, including ChAdOx1, one of the candidates leading the race for a Severe acute respiratory syndrome coronavirus 2 (SARS-CoV-2) vaccine [168]. Interestingly, five registered human clinical trials are exploring the use of EVs as therapeutics against coronavirus disease 2019 (COVID-19) [169,170,171,172,173]. Besides, EV-based vaccines may contribute to the new era of mRNA vaccination. The use of mRNA-based vaccines is revolutionizing the field of rapid development of vaccines that is key for development of vaccines against emerging pathogens and has shown promising results in personalized cancer vaccines via delivery of mRNAs encoding neoantigens of each patient [174,175]. One of the most critical steps in mRNA vaccination is achieving an efficient cytoplasmic delivery to target cells. However, this step still poses challenges that need to be overcome such a toxicity of formulations of reaching target cells in secondary lymphoid organs [176]. EVs and DEVs have not been explored yet as possible carriers in mRNA vaccination although they have previously shown to be excellent platforms for delivery of mRNAs and to efficiently reach secondary lymphoid organs [177].

In most cases, the rationale behind the use of DEVs as tumor vaccines or vaccines for oncogenic pathogens relies on the classical definition of vaccines as inductors of antigen-specific immune responses and adaptive memory. However, the broadening of the concept of vaccination with the appearance of the first generation of trained immunity-based vaccines [178] opens new horizons in exploiting this new arm of the immune system as a new source of therapeutic strategies for immunotherapy and cancer vaccines. Indeed, it needs to be determined to what extent trained immunity is playing a role in the previous examples of DEVs vaccination (e.g., by indirectly enhancing the antigen-specific adaptive response) or in the immunotherapeutic interventions that we discussed, most of which are based on immunomodulatory effects in which trained immunity may have a very important role.

## 4. Conclusions

Extracellular vesicles constitute one of the most efficient mechanisms for intercellular communication. In the context of tumor development, EVs are present in the tumor microenvironment and are secreted by a variety of cell types. On the one side, immune cells produce EVs to fight against tumor progression and metastasis. As illustrated before, DCs, in particular, can secrete EVs to increase the T cell response by enhancing antigen presentation by a diversity of mechanisms (Figure 1A). On the other side, malignant cells produce EVs to escape immune responses and virtually, they can interact with every type of immune cell. Nonetheless, the complexity of these interactions makes it difficult sometimes to define a clear role of these vesicles. In fact, there is still some controversy around the immunosuppressive or immunostimulatory role of TDEVs. As we have mentioned, TDEVs affect myeloid precursor differentiation towards MDSCs, affecting monocytic and DC function (Figure 1B). Besides, they can directly modulate NK and T cell proliferation and apoptosis. Nevertheless, other studies pointed out that TDEVs carry TAAs, enhancing the DC responses against malignant cells. More in vivo studies will shed some light on the understanding of the complex communication net that modulate cancer development. 

The great ability of DEVs as immune modulators and kick-starters of robust antigen-specific T cell responses and NK cell responses have allowed the use of DEVs in different immunotherapeutic settings: both as novel and effective cancer vaccines and cancer immunotherapies (Figure 2). Several in vivo studies as well as clinical trials support the increased efficacy of the use of DEVs as cancer vaccines compared to DC-based vaccines. However, the prophylactic use of DEVs as a novel and versatile vaccine platform against infectious agents, including oncopathogens, is still at its inception. Combining the use of DEVs and emerging technologies such as mRNA vaccination or the exploitation of trained immunity mechanisms will push forward the frontier of tumor vaccination approaches. 

## Figures and Tables

**Figure 1 cancers-12-03558-f001:**
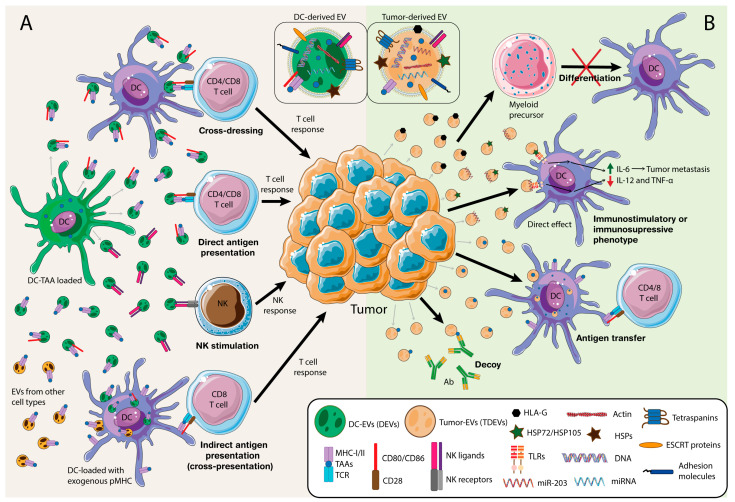
The role of extracellular vesicles (EVs) in dendritic cell (DC)-tumor cell interactions. (**A**) DCs carrying tumor-associated antigens (TAAs) produce Dendritic cells (DCs)-derived EVs (DEVs) that convey a variety of molecules. Particularly, DEVs are characterized by TAA-loaded major histocompatibility complex (MHC)-I/II and co-stimulatory molecules. These vesicles can coat other DCs, a process known as cross-dressing, or can directly interact with T cells, both processes leading to antigen presentation to CD4^+^ or CD8^+^ T cells. DCs, that did or did not contact the tumor before, can also incorporate TAA-loaded-DEVs and mediate antigen presentation to CD8^+^ T cells, called cross-presentation. Moreover, DEVs carry natural killer (NK) ligands on their surface, promoting NK stimulation and helping the immune antitumor response. (**B**) Tumors produce EVs (TDEVs) bearing TAAs, MHC-I/II and many other molecules. These TDEVs can interfere DC differentiation towards myeloid-derived suppressor cells (MDSCs). Moreover, specific molecules in these EVs can be recognized by toll-like receptors (TLRs) on the DC triggering a signaling cascade that ultimately leads to tumor metastasis or downregulates cytokine production. TDEVs bearing TAAs could be uptaken by DCs and are subsequently presented to T cells. Lastly, TDEVs can modulate other immune responses, like sequestering Ab on their surface, acting as decoys instead of coating the malignant cell. HSPs, heat shock proteins; HLA-G, non-classical human leukocyte antigen-G; ESCRT, endosomal sorting complexes required for transport; TCR, T cell receptor; miR-203, microRNA-203; and miRNA, microRNA.

**Figure 2 cancers-12-03558-f002:**
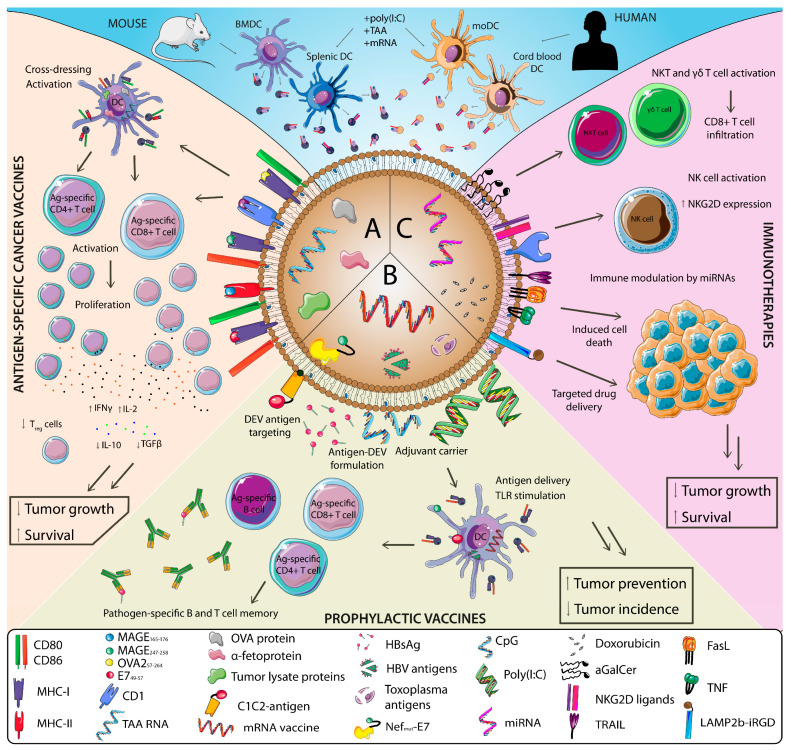
Therapeutic and prophylactic applications of DEVs in cancer. Mice and human DEVs have been used in antitumor studies as therapeutic or prophylactic agents. (**A**) DEV-based cancer vaccines have been designed by loading DEVs with peptide-loaded MHC (pMHC) molecules, tumor proteins and lysates, or mRNAs that encode neoantigens with the goal to mount antigen (Ag)-specific antitumor responses. (**B**) DEVs-based anti-pathogen vaccine platforms for cancer prevention have been designed either by natural-occurring antigen loading to DEVs or by fusing antigens to DEV proteins. Also, their ability as adjuvant carriers and antigen-DEV formulations has been explored to increase vaccine efficacy. In addition, DEVs are a potential platform for mRNA vaccine delivery. (**C**) DEVs have been used as immunotherapeutic agents by exploiting their immune stimulatory properties. DEVs can directly stimulate innate cells such as natural killer T cells (NKT), NK, and γδ T cells or modulate the immune response indirectly by the delivery of miRNAs; or induce tumor cell apoptosis via ligand interaction or targeted delivery of chemotherapeutics. MAGE, melanoma antigen gene; OVA, ovalbumin; E7, human papillomavirus E7 protein; HBV, hepatitis B virus; HBsAg, surface antigen of the HBV; Nef, negative regulatory factor from human immunodeficiency virus; α-GalCer, α-galactosylceramide; poly(I:C), polyinosinic:polycytidylic acid; FasL, Fas ligand; TNF, tumor necrosis factor; LAMP2b–iRGD, lysosome-associated membrane protein 2 (LAMP2b) fused to the integrin-specific peptide iRGD.

**Table 1 cancers-12-03558-t001:** Clinical trials using DEVs performed to date and their main immune and clinical outcomes.

Targeted Tumor Type	Phase of Trial	*n* ^1^	Treatment	Loaded Antigen	Immune Effects	Clinical Outcome	Ref.
NSCLC (stage IIIb and IV)	I	13 (9)	DEVs from moDC	MAGE-A3, -A4, -A10, and MAGE-3DPO4 peptides + CMV and tetanus toxoid peptide (direct or indirect loading)	DTH reactivity against MAGE peptides in 3/9. MAGE-specific T cell responses in 1/3. NK lytic activity in 2/4. CMV responses. Increase of T_regs_ in 2/3.	Well tolerated. Mild adverse events. Stabilization after progression in 2/9.	[163]
Melanoma (stage IIIb and IV)	I	15	DEVs from moDC	MAGE3_168–176_ and MAGE3_247–258_	Specific T cell responses in peripheral blood not detected. One case of tumor infiltration of activated T cells. NK cell number and NKG2D function recovered in 7/14. NKG2D expression in CD8^+^ T cells in 6/14.	No toxicity (mild adverse events). One patient exhibited a partial response.	[41,164]
Colorectal cancer (stage III or IV)	I	40	AEVs + GM-CSF	Contain CEA	DTH response as well as a CEA-specific CTL cell response	Well tolerated. Stabilization in 1 and minor response in 1.	[165]
NSCLC (stage IIIb and IV)	II	26 (22)	DEVs from IFNγ-matured moDC	MAGE-A1, MAGE-A3, NY-ESO-1, Melan-A/MART1, MAGE-A3-DP04 and EBV peptides.	Tumor antigen-specific T cell responses only in 2/8. Increased NKp30-dependent NK cell functions.	Stabilization with continuation of injections in 7. Long-term stabilization in 1. Hepatotoxicity in 1.	[166]

^1^ In parentheses, number of patients that completed therapy; NSCLC, non-small-cell lung carcinoma; CEA, carcinoembryonic antigen; AEVs, ascites-derived EVs; CMV, cytomegalovirus; DTH, delayed-type hypersensitivity; moDC, monocytic-derived DC; EBV, Epstein-Barr virus; CTL, cytotoxic T lymphocytes; IFNϒ, interferon-ϒ.

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
