# Peer review of "Immune Regulation by Dendritic Cell Extracellular Vesicles in Cancer Immunotherapy and Vaccines"

_cancers, 2020, doi:10.3390/cancers12123558_

Round 1
Reviewer 1 Report
This review has outlined the current state of knowledge on the effects of DC-derived EVs on tumour and their role in cancer immunotherapy and vaccines. It covers a broad range of information, including a section on the effects of tumour-derived EVs on immune cells, and is generally well written. There are, however, several points that could be addressed to improve the overall quality of the manuscript:
- My main point for revision is that the review covers a broad subject scope and could benefit from some simplification and streamlining of information. For example, in sections 2.1 and 2.2 the discussion also includes immune cells other than DCs. While all of the information is pertinent, it could perhaps be a bit more focussed to the main purpose of the paper. Alternatively, the use of more subheadings or tables outlining information would be beneficial. For example, a table outlining clinical trials to date and their outcomes would have been beneficial in that section.
- The abstract is written to focus on the role of DC-derived EVs, however, there is also a lengthy discussion of tumour-derived EVs within the paper. I would suggest amending the abstract to better reflect this.
- Line 62: “Extracellular vesicles (EVs) encompass a family of nanovesicles”. Not all EVs are nanovesicles, they also range into the micro size. Please amend.
- Figures 1 and 2 are both excellent summaries of the information that’s been given. However, the figure legends are long and could be reduced quite significantly.
- Line 177-179 describes tumour-derived EVs as promoting “pro-inflammatory responses on monocytes, macrophages and DCs”. However, the pro/anti-inflammatory effects of TDEVs are much more complex than this and they can induce anti-inflammatory phenotypes in all of these cell types also. Indeed, it appears that there are significant effects of time and dose of TDEVs, as well as the tumour type and stage, on the response that they will produce.
Author Response
Replies to Reviewer 1:
This review has outlined the current state of knowledge on the effects of DC-derived EVs on tumour and their role in cancer immunotherapy and vaccines. It covers a broad range of information, including a section on the effects of tumour-derived EVs on immune cells, and is generally well written. There are, however, several points that could be addressed to improve the overall quality of the manuscript:
- My main point for revision is that the review covers a broad subject scope and could benefit from some simplification and streamlining of information. For example, in sections 2.1 and 2.2 the discussion also includes immune cells other than DCs. While all of the information is pertinent, it could perhaps be a bit more focused to the main purpose of the paper. Alternatively, the use of more subheadings or tables outlining information would be beneficial. For example, a table outlining clinical trials to date and their outcomes would have been beneficial in that section.
We agree with reviewer 1 that sections 2.1 and 2.2 might be slightly confusing as we want to add some additional information about other immune cells types in order to give a broader view of the microenvironment around tumor cells. Following his/her suggestion, we have now added several subheadings to clarify the structure of both sections, being as follow:
- The role of EVs in cancer development
2.1. Modulation of antitumor immunity by DCs-derived EVs
2.1.1. DCs-derived EVs (DEVs)
2.1.2. DEVs function
2.1.3. Influence of the adaptive response
2.2. Modulation of the immune response by tumor-derived EVs
2.2.1. Tumor-derived EVs (TDEVs)
2.2.2. TDEVs function on myeloid cells
2.2.3. TDEVs function on DCs
2.2.3. TDEVs function on other immune cells
Besides, we have added a table summarizing the outcomes of the clinical trials performed to date, as suggested.
With think this changes will help to understand the content and help the reader to better follow the content of the review. We thank the reviewer for the great suggestions.
- The abstract is written to focus on the role of DC-derived EVs, however, there is also a lengthy discussion of tumour-derived EVs within the paper. I would suggest amending the abstract to better reflect this.
Following reviewer 1 suggestion we have now added and amended the Abstract to highlight TDEVs. We hope this highlights that the review has dedicated a complete section on this matter.
- Line 62: “Extracellular vesicles (EVs) encompass a family of nanovesicles”. Not all EVs are nanovesicles, they also range into the micro size. Please amend.
We agree with reviewer 1 that EVs include vesicles at the range of micro size, as for example apoptotic vesicles. We apologize for the mistake and have now deleted that, new line 73-74.
- Figures 1 and 2 are both excellent summaries of the information that’s been given. However, the figure legends are long and could be reduced quite significantly.
Figures 1 and 2 contain a great amount of information that has been mentioned along the corresponding sections. Therefore, following suggestion of reviewer 1 as it might be self-explanatory and the details can be found along the text, we have now reduced figure legends to keep the essential to understand the figure. We agree that it will help to follow the flow of the figure and remark the main ideas. We reduced from 263 to 192 words in figure 1 and from 183 to 159 in figure 2.
- Line 177-179 describes tumour-derived EVs as promoting “pro-inflammatory responses on monocytes, macrophages and DCs”. However, the pro/anti-inflammatory effects of TDEVs are much more complex than this and they can induce anti-inflammatory phenotypes in all of these cell types also. Indeed, it appears that there are significant effects of time and dose of TDEVs, as well as the tumour type and stage, on the response that they will produce.
We agree that monocytes, macrophages and DCs have an important function also in inducing anti-inflammatory phenotypes as it is mentioned in later sections. We apologize for the mistake. We have unified the phrase to include both possible sides for these cell types: “TDEVs can alter the microenvironment of the tumor promoting both pro/anti-inflammatory responses on monocytes, macrophages and DCs, as well as anti-inflammatory responses acting on NKs, and Treg, thus modulating angiogenesis, invasion, apoptosis, and metastasis.” Line 206-209.

Reviewer 2 Report
In this manuscript, the authors reviewed the published work on the study of extracellular vesicles from dendritic cells (DEVs) in tumor growth and the potential application of DEVs as tumor vaccines. This is a comprehensive review that thoroughly went through the literature in the field. There are some issues that can be addressed in the revision:
- There are a number of mismatches between the text and the references. Some of the statement may not reflect the original study. Here is just a partial list of the references:
Line 112-114, the authors stated DEVs promote T cells response by direct antigen presentation. But reference [52] only shows an indirect effect via DEV-pulsed DCs pathway.
Line 114-117, the references [36,39,55] talked about the direct effect of DEVs on T cells, not via internalization by DCs.
Line 128-129, there are no co-stimulatory factors mentioned in reference [39].
Line 149-150, reference [80] talked about the effect of exosomes from CD8+ T suppressor on macrophage, not CD8+ cytotoxic T cells. The two types of CD8+ T cells must be clarified.
Line 193-194, there is no evidence to show tumor exosomes or non-tumor exosomes in reference [91].
Line 202-204, reference [96] did not mention M1 macrophage, reference [95] claimed M2 macrophage for their results.
Line 210-213, reference [101] it has to be emphasized that the major role of TDEV is promoting the generation of a myeloid immunosuppressive cell subset, not suppress DC differentiation only.
In line 122-23, activated T cells can recruit EVs in immune synapse. According to the Refs, it’s emphasizing the LFA-1, but not “pMHC-loaded EVs”, the original description is somehow misleading.
- In line 181, “HSP80” could be a typo, please double check.
- In line 391, DEVs are able to bind TLR ligands, here should be “carrying” TLR ligands rather than “bind”.
- For the clinical trials on the DEVs on cancer therapy, it would be more helpful to list them in a table.
Author Response
Replies to Reviewer 2:
In this manuscript, the authors reviewed the published work on the study of extracellular vesicles from dendritic cells (DEVs) in tumor growth and the potential application of DEVs as tumor vaccines. This is a comprehensive review that thoroughly went through the literature in the field. There are some issues that can be addressed in the revision:
- There are a number of mismatches between the text and the references. Some of the statement may not reflect the original study. Here is just a partial list of the references:
Fist, we would like to apologize for the any mistake we included while citing references and thank deeply reviewer 2 for the thorough revision of the manuscript.
Line 112-114, the authors stated DEVs promote T cells response by direct antigen presentation. But reference [52] only shows an indirect effect via DEV-pulsed DCs pathway.
We agree with reviewer 2 that reference Hao et al, 2007 (new [54]) only shows an indirect effect, as DCs are pulsed previously with DEVs and no direct effect of DEVs on T cells is shown. Therefore, this reference has been moved to the following statement where we mentioned: “DEVs can also be internalized by other DCs as a source of exogenous peptide-loaded MHC (pMHC), which may be subsequently presented to naïve, primed or memory T cells.” This phrase will match better with what Hao and colleagues described. New line 134-136.
Line 114-117, the references [36,39,55] talked about the direct effect of DEVs on T cells, not via internalization by DCs.
Reviewer 2 suggests that references [36,39,55] described direct effect of DEVs on T cells. While we agree that references [36,39] were mistaken; reference [55] described DEVs internalization by DCs, and specifically DEVs from mature DCs. Montecalvo et al., 2008 described a mechanism of alloantigens transfer through exosomes to DCs to amplify generation of donor-reactive T cells following transplantation. We have included references Admyre et al, 2006 and Segura et al, 2005 [36,39] in a phrase more accordingly to what the original article described. Being as followed: “EVs from an antigen-loaded DC bear tumor antigens that may promote CD4+ and CD8+ T cell responses by direct antigen presentation, which leads to tumor growth suppression [35,36,52] with increased efficiency in the case of mature DC-derived EVs [39]. DEVs can also be internalized by other DCs as a source of exogenous peptide-loaded MHC (pMHC), which may be subsequently presented to naïve, primed or memory T cells [43–45,53–55].” New line 136-137
Line 128-129, there are no co-stimulatory factors mentioned in reference [39].
We disagree with reviewer 2 in this point as showed by Segura et al, 2005; they said: “ Proteomic and biochemical analyses revealed that mature exosomes are enriched in MHC class II, B7.2, intercellular adhesion molecule 1 (ICAM-1), and bear little milk-fat globule–epidermal growth factor–factor VIII (MFG-E8) as compared with immature exosomes”, where B7.2 is CD80 a co-stimulatory molecule that binds CD28 during an immune synapse. But, in order to soften the statement and clarify it we have also added ICAM-1 as it is also an important molecule for T cell activation which is also present in these vesicles. Besides, we agree with reviewer 2 that as Segura and colleagues described while CD80 might be dispensable, ICAM-1 and MHC-II are required for T cell priming. New line 148-149.
Line 149-150, reference [80] talked about the effect of exosomes from CD8+ T suppressor on macrophage, not CD8+ cytotoxic T cells. The two types of CD8+ T cells must be clarified.
Following suggestion from reviewer 2 we have now clarified that the T cells described by Nazimek et al, 2015 are CD8+ T cell suppressor cells(Ts). New line 172-173.
Line 193-194, there is no evidence to show tumor exosomes or non-tumor exosomes in reference [91].
Bretz et al, 2013 described the effect of exosomes isolated from various body fluids on monocytic cell differentiation. One of these sources was malignant ascites of ovarian cancer patients. While we agree that this original article is working in a more ex vivo experimental set up and from malignant and non-malignant sources, Bretz and colleagues described an important observation that would be relevant for the designing of immunosuppressive treatments for cancer. Although, we agree that the observations were not only characteristic of cancerous exosomes. We have left the word EVs as a more general term to refer to this particular study. New line 221.
Line 202-204, reference [96] did not mention M1 macrophage, reference [95] claimed M2 macrophage for their results.
We apologize for the mistake as Wu et al, 2016 ([95], new [96]) do claim a more M2 phenotype in their conclusion, that might be confusing along the text as they claim that their macrophages acquire a proinflammatory phenotype after TDEVs treatment which increases tumor cell growth, migration and invasion, while M-2 are more anti-inflammatory. We have changed the statement to the neutral term “macrophages” as Wu et al, 2016 work with the cell line THP-1 cells and they do claim an increase of proinflammatory cytokines through NFkB. Result: “Additionally, macrophages can produce inflammatory cytokines, including IL-6, IL-8, IL-1β, CCL2, TNF-α and G-CSF, also via NFκB and TLR-dependent by TDEVs stimulation in breast [95] and gastric cancer [96]; the later with increasing tumor cell growth, migration and invasion [96]. Furthermore, M2-polarization, which is more pro-tumorigenic, can be controlled by EVs derived from hepatocellular carcinoma (HCC) cells though SALL4/miRNA146a axis [97].” New line 232-235.
Line 210-213, reference [101] it has to be emphasized that the major role of TDEV is promoting the generation of a myeloid immunosuppressive cell subset, not suppress DC differentiation only.
We agree that MDSCs generation should be emphasized as it has been done in figure 1 legend. TDEVs not only suppress myeloid cells differentiation but also they do so as they induce the accumulation of MDSCs around tumor cells. We have amended that: “For instance, it has been described that TDEVs from melanoma or colorectal carcinoma cells impair CD14+ human monocyte differentiation into DCs, leading to MDSCs generation creating an immunotolerant environment.” New line 244-245.
In line 122-23, activated T cells can recruit EVs in immune synapse. According to the Refs, it’s emphasizing the LFA-1, but not “pMHC-loaded EVs”, the original description is somehow misleading.
We agree on this point. Hence, we have amended the sentence including the fact that the process is described as dependent on LFA-1 according to the cited paper. New line 142-143.
- In line 181, “HSP80” could be a typo, please double check.
We have revised the original article by Andre et al, 2002 [82] and they do mention HSC70 and HSP80, no typo error. But, as HSP80 results were not shown and were only mentioned by Andre and colleagues, we have decided to remove HSP80. New line 209. Moreover, that led us to double check any typo for DEVs, we have now changed HSC80 and HSP90 for HSC73 and HSP84, remarking the specific chaperone described by Théry et al, 2001, instead of the chaperone family they belonged. New line 119.
- In line 391, DEVs are able to bind TLR ligands, here should be “carrying” TLR ligands rather than “bind”.
We have changed the word following the suggestion. The new phrase (now in line 429) is as follows: “DEVs are able to carry TLR ligands”.
- For the clinical trials on the DEVs on cancer therapy, it would be more helpful to list them in a table.
We thank Reviewer 2 for the suggestion. We agree that a table outlining the main outcomes of clinicals trials performed up to date would help the reader to better get the main findings of these trials and therefore we have added a table including them.
